# How structured cultural changes can reduce cesarean section rate in a Danish tertiary hospital

Eva Rydahl[1]*, Kamilla G. Nielsen[2], Ole Olsen[3], Amalie L. Henningsen[4], Helle Johnsen[5]

1 Department of Midwifery, University College Copenhagen, Denmark, 2 Hospital Southern Jutland, Aabenraa, Denmark, 3 University of Copenhagen, Copenhagen, Denmark, 4 University of Southern Denmark, Odense, Denmark, 5 Department of Midwifery, University College Copenhagen, Denmark

* evry@kp.dk

## Abstract

### Background

Cesarean section is rising globally, with the risk of over-use and subsequent iatrogenic consequences for child and maternal health and future pregnancies. This study aims to evaluate the impact of a 12-step initiative implemented by a Danish tertiary hospital, targeting organisational structures, healthcare personnel, and the birthing population, on reducing cesarean section rates, compared to control hospitals that did not adopt a similar approach.

### Methods

A national retrospective register-based cohort study comparing the index hospital with two control hospitals, five years before implementation until eight years after the initiation of the strategy. Interrupted Time Series Analyses are performed with and without the control group. The study was conducted in a Danish tertiary hospital, Hospital Southern Jutland. A sample of term births in Denmark either at the index hospital (n = 21,232) or at two control hospitals (n = 46,417) from 2003 to 2017. Primary outcome was Cesarean section. Secondary outcomes were severe maternal- and perinatal complications.

### Results

During implementation (2008−2017), the cesarean rate decreased at the index hospital from 21.1% to 12.0% (−0.87% annually, p < 0.001). There was no significant change in instrumental birth, uterine ruptures, neonatal intensive care unit admission, or fetal death. The rate of Apgar Score <7/ 5 minutes levelled off after a rising trend (p = 0.009). Both the index hospital and controls had a decline in cesarean rates in the intervention period (−0.87% vs. −0.12% annually), which corresponds to an

**Data availability statement:** All data are contained within the manuscript, in Supporting Information or available at a repository at Zenodo: URL: https://doi.org/10.5281/zeno-do.16910955 / DOI 10.5281/zenodo.16910954.

**Funding:** The author(s) received no specific funding for this work.

**Competing interests:** The Authors have declared that no competing interest exist.

extra 0,75% annual reduction at the index hospital compared to the control hospitals (p < 0.001).

## Conclusion

Implementing a multi-component initiative to reduce cesarean sections has demonstrated both effectiveness and clinical significance. If such a strategic implementation can be conducted elsewhere, it could yield substantial benefits for maternal and fetal health.

## Introduction

Globally cesarean section (CS) rates are rising, and projections by the World Health Organization (WHO) suggest that 28.5% of births will be by CS by 2030 [1]. While CS can be a lifesaving intervention, the WHO generally considers rates below 10.0% to indicate inadequate access to essential obstetric care [2]. Nevertheless, CS rates beyond a certain threshold should raise concerns about the potential iatrogenic consequences for the health of mothers, their off-springs, and future pregnancies [3]. Increasing CS rates have not been accompanied by significant maternal and perinatal benefits, and in 2015 the WHO stated that at a populational level, CS-rates above 10.0% do not lead to reductions in maternal and new-born mortality rates [3].

An increase in the proportion of nulliparous women and older or more obese women may be associated with the rise in CS rates [4]. Still, these changes are unlikely to solely explain the accelerating CS rates and the variations observed between countries [4]. Among the 34 members of The Organization for Economic Cooperation and Development (OECD), the CS rate varies from 14.8% to 53.1% (2019)[5], suggesting considerable variation in otherwise comparable countries. OECD finds these variations attributed to various factors, such as CS availability, obstetric training, financial incentives, and liability concerns. Further, preferences among women may contribute to increased CS-rate, but these preferences are likely linked to the organisation of the maternity care system and cultural attitudes towards labor and birth [5].

High rates of CS cause concern as the surgery is associated with both short- and long-term risks. Maternal adverse effects may be organ injury, infection, anaesthesia complications, and thromboembolic disease [6]. In subsequent pregnancies, the risk of uterine rupture, placenta accrete, -previa, ectopic pregnancy, infertility, hysterectomy, and intraabdominal adhesion increase [6–11]. Further, CS may lead to adverse birth experiences and promote poorer parenting behaviours [12]. The newborn has an increased risk of neonatal respiratory distress, asthma and obesity [6,9,10].

Various strategies – both non-clinical and clinical – have been suggested to reduce CS rates [13]. Non-clinical strategies could be, e.g., health promotion, health education and financial management, where clinical strategies could include practice guidelines, training of staff, feedback and childbirth training [14,15]. In 2014, The American College of Obstetricians and Gynaecologists published a program to prevent nulliparous women from having a CS [16]. Despite this national initiative, the CS rate has continued to increase in the United States (US) and is currently at 32,4%

in 2024 [17]. More recent studies suggest a multi-component approach, which includes factors such as health system organisation, culture, and finance to develop strategies for reducing CS rates [13,14]. Likewise, WHO has proposed a broad approach targeting the birthing population, the health care providers, and the organisation, including childbirth and psycho-education, implementing evidence-based clinical guidelines and a mandatory second opinion before performing any CS. The use of a collaborative midwifery-obstetrician model of care is recommended [4]. However, despite initiatives aimed at reducing CS rates, the effectiveness of such interventions has been limited so far [14,18].

### The CS reduction initiative in Denmark

Denmark is a high-income country with free access to healthcare. Normal pregnancies, and childbirth are managed independently by authorized midwives. In case of complications, obstetricians take over responsibility, working in close collaboration with the midwife. The CS rate is relatively stable at approximately 20.0%, fluctuating between 19.0% and 22.0% [19]. Although low in international comparison, the rates are considerably higher than the ideal 10.0–15.0% rate suggested by WHO [3]. In 2008, an urban tertiary hospital in the Southern Region of Denmark, Hospital Southern Jutland (HSJ), introduced a systematic strategy to lower the CS rate without compromising maternal and neonatal health [20]. The motivator for the initiative was an observed increase in the CS rate. A department manager inspired by in-hospital birth clinics in Sweden was engaged to support the cultural and structural changes [21]. The Swedish clinics were known for high maternal satisfaction, personalised care, informed decision-making, and collaborative patient-provider communication [21].

The initiative at HSJ included cultural, structural, and clinical changes. First, common values were agreed upon between midwives, obstetricians and other staff; that vaginal birth and a good birthing experience are important, that obstetric knowledge is essential when safely reducing intervention rates, and that staff who feel competent in both the uncomplicated vaginal birth as well as emergencies will increase birthing women's perceptions of receiving appropriate treatment. This was followed by a step-by-step implementation of different actions aiming to perform as few interventions as possible without compromising safety and maternal satisfaction. Among interventions, a 24-hour midwife coordinator function was instigated. A morning conference was implemented with daily teaching and mutual reflections over CSs the previous day. Coaching and supervision groups were made accessible to staff. A new "Birth planning clinic" addressed women's childbirth fears and provided evaluation after obstetric emergencies (e.g., acute CS). New and optimised solutions were implemented for homestays for women with pregnancy-related problems as an alternative to in-hospital stays. For new parents, standard postpartum and post-surgical consultations were offered.

Clinical alterations included new training sessions to help prevent cesarean sections. This involved intrapartum ultrasound, manual rotation, rebozo for labor dystocia, and training for obstetricians to collaborate with midwives during normal birth. As a result, obstetricians were increasingly invited into the birthing rooms to be part of the decision-making team. Collaborative obstetric team training for emergencies was also part of these changes. Furthermore, to enhance transparency, close data monitoring of all births with visual monthly reports was introduced and evaluated regularly, and the statement about valuing vaginal birth was made public to women, their partners, and the local community. In total, 12 initiatives were implemented.

In S1 file the multi-component initiative is visualised and described in detail.

This study aims to analyse the impact of a structured strategy to reduce CS rate implemented stepwise at HSJ from 2008 to 2017. The research question is if the intervention reduced the CS pattern without compromising fetal and maternal health. Further, if HSJ can show a reduction in the risk of CS, that is significantly different from similar hospitals.

## Materials and methods

The method section is separated into sections on the study design and data sources, followed by sections on the population-, the intervention-, the control group- and the outcome of interest. This is followed by a statistical analysis, an ethics statement, and a section on patient and public involvement.

## Study design and data sources

The study uses prospectively collected data in a retrospective cohort design. For this purpose, data from the Danish Medical Birth Registry were sampled with additional Danish administrative registries to add further relevant patient data (e.g., education and citizenship). The dataset holds information on all Danish pregnancies from 22 gestational weeks and onwards, through childbirth until hospital discharge. In Denmark, all citizens hold a civil or temporary registration number. Thereby it is possible to register all contacts with healthcare providers [22]. Denmark has free access to hospitals during childbirth, which is why it is most likely that undocumented migrants giving birth are registered too [23]. Data are hosted in Statistics Denmark (MIPAC 705026). Data are based on routinely collected and aggregated data and handled according to European General Data Protection Regulation [24].

## The population of interest

HSJ is a small/medium-sized Danish tertiary hospital located in an urban area of Denmark. The unit has six birthing rooms available and is staffed 24/7 by midwives, healthcare assistants and specialist obstetricians. Annually, the hospital has an approximate total of 1550 births. The labor ward has direct access to an intensive care unit and a neonatal intensive care unit (NICU).

We measured five years before the onset of the new strategy (2003–2007) and nine years after (2008–2017), including births from 1 January 2003 until 31 December 2017. During this period, a total of 938.365 pregnant people gave birth in Denmark. In this article, we use the term 'women' for all pregnant people. This choice is made for consistency within the study. Only births eligible for the multi-component initiative were included. Thereby, preterm births <37 + 0 gestational weeks were excluded (N = 63,747), as well as births with no registered gestational age (N = 17,518). The majority had low, or no birth weight registered pointing to late abortions. To avoid confounding by indication, cases where CS was inevitable were excluded, such as placenta previa (N = 4310) and transverse/oblique fetal lie (N = 1612). Further, all pregnancies complicated by diabetes were excluded from the total sample, as the index hospital HSJ transfers diabetic women with insulin-dependent diabetes to a regional hospital with expertise (N = 26,693). At HSJ, trial of labor after previous CS was restricted to two previous CS, which is why pregnancies with more than two previous CS were excluded (N = 1076). The total population after restriction included 823,409 births in Denmark. Among these, 21,232 gave birth at HSJ. (Fig 1, Flowchart)

## The intervention of interest

Between 2008 and 2017, HSJ developed and implemented a strategic initiative aimed at reducing the CS rate. The initiative included 12 different components, like, e.g., team training, audit feedback, and birth plans for women after CS or traumatic births (S1 File). The rollout began in 2008, coinciding with the appointment of a new head of department who was specifically tasked with leading the strategy. For this analysis, 2008 is considered the year of introduction.

## The control group(s)

This study will include a "single group" analysis (ITSA) and an analysis with a control group (CITSA). In the single-group analysis, the intervention period (2008–2017) in HSJ acts like HSJ's control group/counterfactual for the pre-intervention period (2003–2007). As a result, the trend in outcome in the preintervention period (e.g., CS) is compared to the trend in the intervention period. The assumption is that the population characteristics change slowly over time (e.g., maternal age, obesity rates, parity). Thereby, the pre-intervention and the intervention population characteristics presumably are unaltered [25]. However, this approach cannot exclude confounding due to co-interventions or other events occurring around the time of the intervention (e.g., a national strategy for reducing CS rates). The latter problem can be handled by performing analyses which compares the development over time at HSJ with the development over time in the control groups

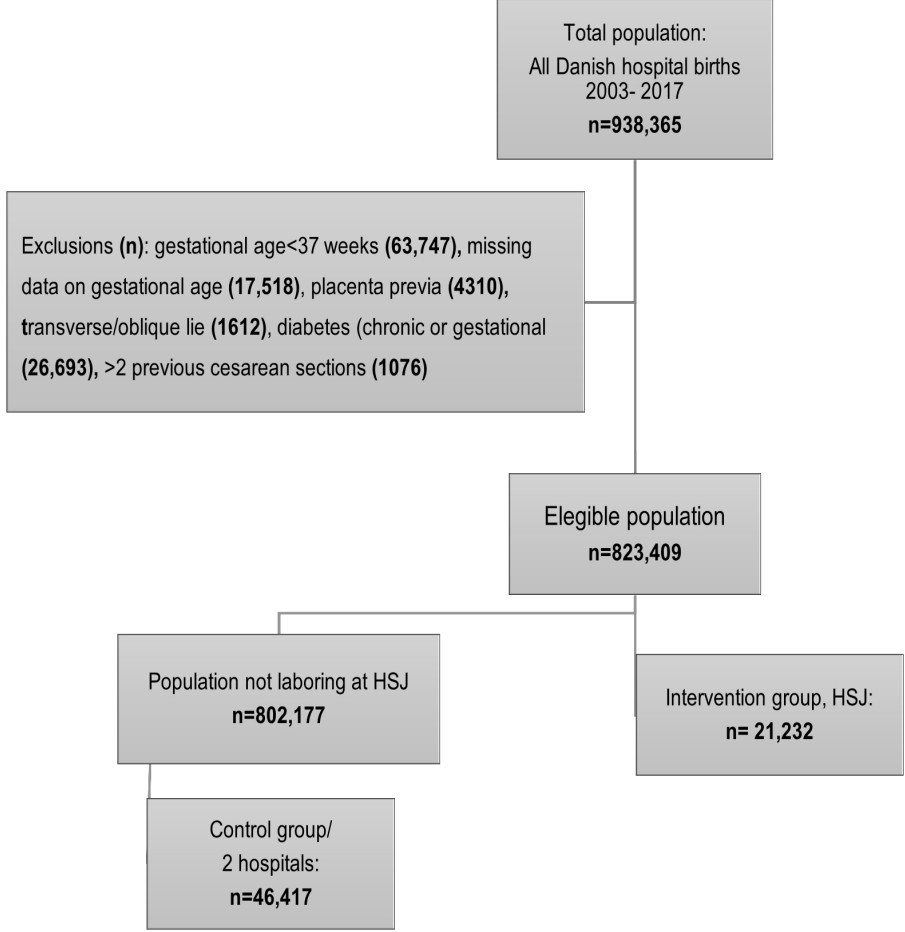

**Fig 1. Flow chart of study participants.**

(CITSA) [26]. When the index hospital is compared to the development over time at the control hospitals in the same time interval, it will be possible to observe whether time-dependent factors affect both populations [25–27]. To strengthen the analyses, two hypotheses were tested: First, if the trend over time before the intervention was initiated differed from the trend observed after the intervention began. The null hypothesis posited that the trend would remain unchanged. Secondly, the trend over time following the intervention at the index hospital differed from the trend among the control group. The null hypothesis suggested that the trend over time would be the same for both the index hospital and the controls.

The control group of hospitals that resemble HSJ was selected incrementally under the following criteria:

- The labor ward has existed throughout the years of interest; Twenty Danish hospitals existed from 2003–2017 (n = 20).

- Approximately identical annual births as HSJ; Four units had a mean variation of births not exceeding +/- 500 annually compared to HSJ (n = 4).

- Include the same level of care as HSJ: access to immediate operative support and neonatal intensive care (n = 3).

- The control unit should not be in the same region to avoid the risk of local policies and initiatives (n = 1). One hospital (Esbjerg) is excluded.

Two tertiary labor wards in urban areas met the inclusion criteria (Holbaek & Hjoerring). The control group was generated for robustness by merging these two units into one control group (N = 46,417). A second group, including all other Danish hospitals except HSJ, mirrors Denmark's overall trends and policies over time (N = 802,177) ([Fig 1]).

## The outcomes of interest

All outcomes are presented as dichotomous outcomes (yes/no). The primary outcome is CS. However, CS is stratified into the timing of CS, that is "before onset of labor" or "intra-partum". Before onset of labor includes "planned CS" and "acute CS before onset of labor". For intra-partum CS, women must be in labor before a planned CS or CS caused by birth complications. Furthermore, CS rates are stratified by parity (nulliparous or multiparous). In the single-group analysis, secondary outcomes were assessed. For newborns, these included: Apgar Score <7/ 5 minutes; admittance to NICU >24 hours; and fetal death. For the mothers, secondary outcomes included instrumental birth (forceps and/or vacuum extraction), and uterine rupture (both partial and complete).

## Statistical analysis

Baseline characteristics for the groups were performed by combining the five years before implementing the strategy (2003–2007). Means and standard deviations (SD) are presented for continuous variables if normally distributed; otherwise, using median and interquartile range (1–3. quartile). For dichotomous variables, percentages are used.

Baseline variables included parameters that may impact the risk of CS: mean maternal age, advanced maternal age (≥35 years), parity (nullipara, multipara), previous CS, level of education (low, medium, high), citizenship (Danish or not), severe preeclampsia (blood pressure ≥160/110, proteinuria and/or indicators in blood tests or subjective symptoms), "medical diseases" relevant for pregnancy and childbirth registered by the general practitioner or obstetrician, mean Body Mass Index (BMI), pre-pregnancy obesity (BMI ≥ 30), smoking after 1. trimester, breech presentation, mean birthweight, birthweight ≥4000 gr, and fetal death (after 37 + 0 gestational week).

Analyses were performed using single- and control-group Interrupted Time Series Analysis (ITSA and CITSA). The year of childbirth served as the independent variable and was separated into half-years (n = 30). The time was sequenced into a pre-intervention period (2003–2007; 10 observations) and a post-intervention period (2008–2017; 20 observations). The ITSA model fitted an ordinary least square line (OLS) pre-and post-intervention. We used Newey West to calculate standard errors and the Cumby-Huizinga test for autocorrelation [27]. If lags were present, the analyses were modified accordingly.

Visual descriptive figures present the difference between hospitals. Outcomes are presented in tables according to analytic design (ITSA or CITSA), including the annual slopes ((with a 95% Confidence interval)(95% CI) before and after the HSJ implementation. P-values represent the statistical difference between the slopes pre- and postintervention for HSJ and the difference in slopes between HSJ and control.

The potential impact of missing data depends largely on why these are missing [28]. The current dataset can generate missing data randomly when a health care provider forgets to register a procedure or a condition. In such cases, the degree of missing data cannot be determined, but it may lead to underestimating events or conditions. However, economic refunds for healthcare activities incite the Danish hospitals to have optimal registration [22]. Other variables are mandatory to report (e.g., Apgar score). For these variables, the extent of missing registrations is measured. We included only variables where at least 95% were coded. This standard excluded valuable outcome variables such as postpartum haemorrhage, where coding before 2011 was inadequate. The variable with the highest frequency of missing data was BMI, with 10,0% missing. The lack of data was mainly due to an unsystematic registration in 2003. By restricting BMI data from 2004–2017 in the baseline tables, missing data on maternal BMI decreased to 3.8%. For NICU admittance, an organisational change at HSJ technically merged NICU admittance with the ordinary hospital stays for mother and baby from 2013 onwards, which is why NICU cases cannot be separated. Thus, we restricted the post-intervention period regarding the NICU to 2008–2012. Data on breastfeeding were available from an external health visitor dataset; however,

due to a high degree of unsystematic missing data, they were excluded from all analyses. In S1 Table & S1–S3 Figs, extra information on absolute numbers, parity, Robson classification and change in baseline characteristics over time is presented (see Results section). Repository datasets are accessible at https://doi.org/10.5281/zenodo.16910955

STATA 17.0 (StataCorp 2021). Stata statistical software was used for data management and analyses. All reported p-values are two-sided and with alpha = 0.05 as the statistical level. For cohort reporting, the STROBE guidelines were used [29].

### Ethics statement

This study utilizes retrospectively collected data hosted by Statistics Denmark. All data were fully anonymized before being made available to the researchers. The data are publicly presented in an aggregated format, not requiring patient consent. The dataset used for the analysis is hosted by Statistics Denmark (project ID: MIPAC 705026) and is administered by University College Copenhagen. The Institutional Review Board at University College Copenhagen approved the study (approval no. FK 2023/03).

### Patient and public involvement

As this is an observational study on routinely collected data, there is no direct patient involvement in the study. To enhance public involvement, we contacted the Danish Non-Governmental Organisation "Parenthood and Childbirth", a democratic member-driven organisation working to promote women's and partners' influence on pregnancy and childbirth. The group was involved before any protocol was outlined, and outcome measures were developed in collaboration. Some suggested outcomes were relevant but could not be monitored: breastfeeding duration was inadequately recorded, and maternal satisfaction was not captured in the Danish Medical Birth Registry. Dissemination of the study entailed "Parenthood and Childbirth" using their socially available platforms to promote the study and to discuss the results.

## Results

The key outcome of this multi-component initiative was a sustained reduction in the CS rate from 21.1% to 12.0%, without compromising maternal or fetal health. While both HSJ and the control hospital showed declining CS rates after 2008, the intervention yielded an additional annual reduction of 0.75% at HSJ. The following section presents descriptive statistics, followed by time series analyses, both single and with a control group.

### Descriptive results

After applying exclusion criteria, the final eligible population consisted of 823,456 individuals. Among these, 21,232 gave birth at HSJ, while 46,417 gave birth at the two control hospitals. The Danish CS trend includes all births excluding HSJ, which is n = 802,177 (S1 Table).

### Baseline characteristics

Table 1 shows baseline characteristics for the index population (HSJ) compared to controls at baseline. We used the pre-intervention period (2003–2007) to assess baseline.

HSJ and control differ marginally. At HSJ, the population has 3.2% fewer nulliparous and 1.3% more with a previous CS, 1.8% more are not Danish, 1.9% less are diagnosed with medical diseases, and 1.2% more have a BMI > 30. There is a difference in fetal death at baseline (0.31% vs. 0.14%). HSJ and control represent urban and smaller Danish labor wards heterogeneous to the overall Danish characteristics. At baseline, the general Danish population include more multiparous, more aged >35, more women with higher education, fewer with obesity and fewer smokers than the HSJ and the control population (Table 1).

**Table 1.** Baseline characteristics for HSJ, control and Denmark (without HSJ) in the preintervention period, N = 289.045.

| Baseline characteristics, pre-intervention period, 2003–2007 | | | |
|---|---|---|---|
| *Characteristic* | HSJ<br>n = 7,129 | Control<br>n = 19,215 | Denmark<br>n = 281,916 |
| **Age, mean ±SD** | 29.5 ± 4.9 | 29.3 ± 4.9 | 30.1 ± 4.7 |
| **Age, 35+, n (%)** | 1075 (15.1) | 2786 (14.5) | 49,754 (17.7) |
| **Nulliparous, n (%)** | 4330 (60.7) | 11,376 (63.9) | 157,167 (56.5) |
| **Multiparous, n (%)** | 2799 (39.3) | 6435 (36.1) | 121,221 (43.5) |
| **Previous cesarean, n (%)** | 768 (17.4) | 1828 (16.1) | 26,182 (16.7) |
| **Educational level, n (%) [a]** | | | |
| Low | 1019 (14.9) | 2758 (14.9) | 32,783 (12.2) |
| Medium | 3407 (49.9) | 9437 (51.0) | 117,267 (43.7) |
| High | 2422 (35.4) | 6316 (34.1) | 118,533 (44.1) |
| **Not Danish Citizenship, n (%)** | 797 (11.3) | 1811 (9.5) | 35,935 (12.9) |
| **Severe preeclampsia, n (%)** | 12 (0.17) | 66 (0.34) | 839 (0.30) |
| **Medical diseases, n (%)[b]** | 21 (0.29) | 421 (2.2) | 5074 (1.8) |
| **BMI, median (1.-3. quartile) [c]** | 23.9 (21.5-27.6) | 23.8 (21.3-27.4) | 23.0 (20.8-26.1) |
| **Obese, BMI > 30, n (%) [c]** | 766 (15.0) | 2047 (13.8) | 22,282 (10.6) |
| **Smoking > 1. trimester, n (%)** | 1316 (18.6) | 3966 (20.9) | 39,835 (14.4) |
| **Breech presentation, n (%)** | 305 (4.3) | 766 (4.0) | 11,531 (4.1) |
| **Birthweight gr., mean ±SD** | 3531 ± 521 | 3543 ± 505 | 3549 ± 498 |
| **Birthweight >4000 gr, n (%)** | 1254 (19.1) | 3282 (18.4) | 48,327 (18.4) |
| **Fetal death, n (%) [d]** | 22 (0.31) | 32 (0.17) | 509 (0.18) |

[a]. Educational level: Low = basic education; Middel = secondary education/ skilled worker/short higher education; High = medium higher/bachelor/long higher/ph.d.

[b]. Medical disease that may impact pregnancy and childbirth. The general practitioner or obstetrician scored them.

[c]. Only data from 2004–2007. HSJ n = 5701/ control n = 15.708/ Denmark n = 210.889

[d]. In available data, prelabor and intrapartum deaths cannot be distinguished. However, in Denmark, almost all deaths happen before the onset of labor.

## Total CS during 2003–2017

Denmark CS rate remained relatively stable, fluctuating between 19.0% and 22.0% from 2003 onward 2017. However, during the preintervention period from 2007 to 2013, a gradual increase was observed, with the rate rising from about 18% to 20%. The CS rate rose at both HSJ and the control hospitals during this period. After 2008, the control group increased their CS rate to approximately 22–23%, whereas HSJ decreased to 12.0% annually (2017).

Fig 2 shows the overall CS rate at HSJ, the control hospitals and Denmark (excluded HSJ).

For absolute numbers and percentages, see S1 Table. Furthermore, S1 Fig. stratify CS rates by parity and timing (before onset of labor vs. intra partum). As extra information, S2 Fig. presents CS into the Ten Group Classifications System [3].

## Time series analysis

Assuming gradual changes in population characteristics over time, ITSA does not inherently adjust for potential confounders. Albeit including a control group (CITSA) account for confounding by co-interventions (e.g., national policy changes on CS rates) [23]. Population-level trends in nulliparity, maternal age > 35 years, BMI > 30, and smoking are shown in S3 Fig., which reveal the same time trends observed between HSJ and the control hospitals.

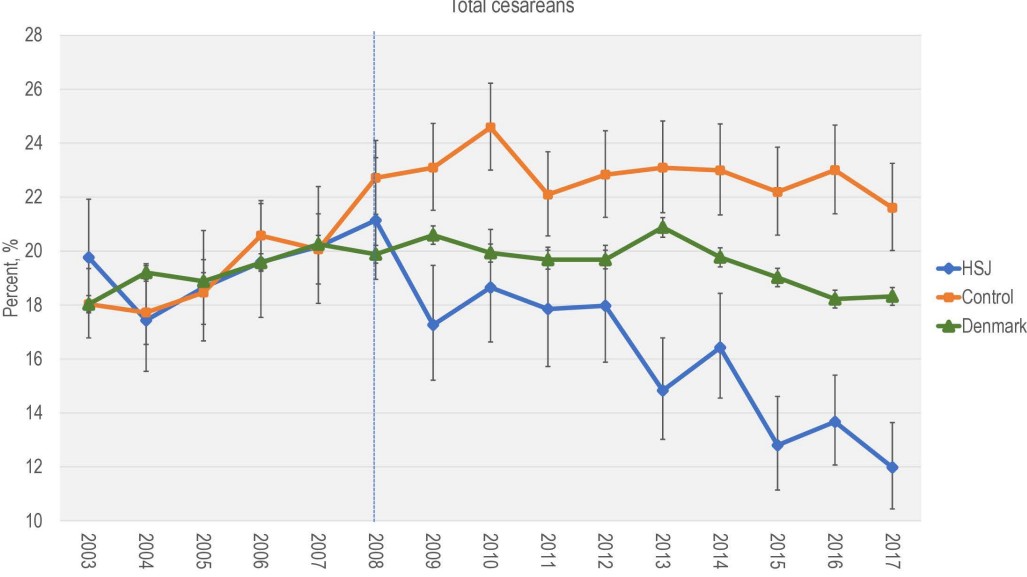

**Fig 2. Cesarean section rate 2003-2017 among HSJ, control and Denmark.** Presented in per cent (%) with 95% CI. Dotted line: time for initiation of intervention.

## Single analyses: HSJ pre-period versus the intervention period

Single-group analyses statistically calculate if the trend at HSJ differs between the pre- and the intervention period. Table 2 presents outcomes for the single-group ITSA with a pre-intervention trend, an intervention trend and a significance test to determine if the difference in trends is significant (Table 2).

The CS rates increased in the preintervention period for total CS (for example, +0.20% annually), nulliparous, and CS before onset of labor. After introducing the strategy, these groups have a statistically significant decrease in CS

**Table 2. Timeseries analyses HSJ trend before- and during intervention.**

|  | Pre-intervention trend | Trend during intervention | Difference in trends |
|---|---|---|---|
| **Maternal outcome** | % per year (95% CI) | % per year (95% CI) | P-value |
| Total cesarean | 0.20 (−0.36 to 0.77) | −0.87 (−1.16 to −0.56) | 0.002 |
| Total cesarean, nullipara | 0.28 (- 0.29 to 0.85) | −0.35 (−0.51 to −0.19) | 0.04 |
| Total cesarean, multipara | −0.08 (- 0.41 to 0.25) | −0.51 (−0.71 to −0.31) | 0.03 |
| Cesarean before onset of labor | 0.09 (−0.42 to 0.60) | −0.60 (−0.77 to −0.43) | 0.02 |
| Cesarean intra partum | 0.12 (−0.22 to 0.47) | −0.27 (−0.49- to −0.05) | 0.07 |
| Uterine rupture | 0.013 (−0.046 to 0.049) | 0.012 (−0.013 to 0.015) | 1.00 |
| Instrumental birth | −0.27 (−0.76 to 0.22) | −0.22 (−0.46 to 0.00) | 0.06 |
| **Perinatal outcome** |  |  |  |
| Apgar score<7/ 5 minutes | 0.17 (0.07 to 0.26) | 0.02 (−0.04 to 0.08) | 0.009 |
| NICU> 24 hours* | 0.40 (0.002 to 0.79) | 0.24 (−0.30 to 0.79) | 0.43 |
| Fetal death | −0.08 (−0.15 to 0.005) | −0.02 (−0.61 to 0.004) | 0.15 |

* NICU is only measured until 2012 inclusive, as an organisational merging of units makes it impossible to distinguish NICU cases in dataset

rates, which reflects a negative trend (e.g., overall CS rate decreased post-intervention by −0.87% annually, (p = 0.002)). Of other significant findings, multiparous had a slightly decreasing trend in the pre-intervention period, but this trend decreased more rapidly after the HSJ intervention in 2008 (p = 0.03).

For the secondary outcomes, only Apgar <7/5 minutes changed significantly. After an upgoing trend pre-intervention (+0.17% annually) Apgar <7/5 had a steady state post-intervention (+0.02%) (p = 0.009). Uterine rupture (p = 1,0), instrumental delivery (p = 0.06), admittance to NICU (p = 0.43) and risk of fetal death (p = 0.15) remained unaltered.

**Multiple analyses: HSJ versus control group**

In the CITSA, we tested if trends regarding CS rates differed from HSJ and the control group. Fig 3 shows CITSA with a pre-intervention trend and a post-intervention trend.

Table 3 presents trends at HSJ and control in the post-intervention period and a significance test to determine the difference. See S1 Fig. for visual inspection.

The overall CS trend did not differ in the pre-intervention period between HSJ and control (p = 0.10). In the post-intervention period, an annual decrease in CS rates is noted at HSJ (−0.87%; 95% CI [−1.16 to −0.56]) and the control group (−0.12%; 95% CI [−0.29 to 0.06]) (p < 0.001). The overall decline suggests a growing national awareness in Denmark of the need to stabilize or reduce CS rates, even without formal policy changes. Both HSJ and control managed to change the CS rates in the pre-intervention period to a decrease in the post-intervention period, but HSJ with an annual extra −0.75% (95% CI [−1.1 to −0.41, p < 0.001). Stratified by parity, nullipara has significantly different trends after the intervention. HSJ reduced the risk of CS annually by −0.35% (95% CI [−0.51 to −0.19]), whereas there is an increase in the control hospitals by +0.53% (95% CI [0.42 to 0.63]) (p < 0.001). Looking at CS before the onset of labor, HSJ has an

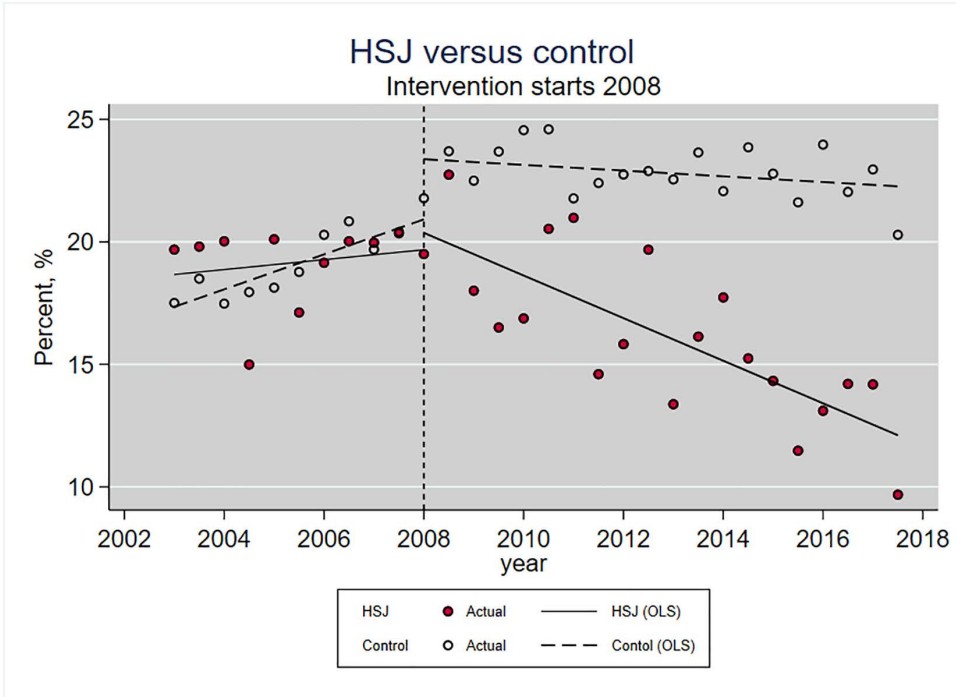

**Fig 3. Cesarean section.** HSJ versus control hospitals, CITSA. Dots: biannual CS rates at HSJ (red) and control (white). Ordinary least square line (OLS) for each hospital before and after the intervention. The dotted line (2008) represents the time of implementation.

**Table 3. Time series comparing trends at HSJ and control hospitals in the pre-intervention and intervention period. Significance test determine if trends differ between groups.**

| Maternal outcome | Difference between groups | | | | |
|---|---|---|---|---|---|
| | Pre-intervention | Post difference between groups | | Post-intervention trend | |
| | P-value | HSJ % per year (95% CI) | Control hospitals % per year (95% CI) | Difference HSJ/control % per year (95% CI) | P-value |
| Total Cesarean | 0.10 | −0.87 (−1.16 to −0.56) | −0.12 (−0.29 to 0.06) | −0.75 (−1.1 to −0.41) | <0.001 |
| Total cesarean, nullipara | 0.02 | −0.35 (−0.51 to −0.19) | 0.53 (0.42 to 0.63) | −0.87 (−1.06 to −0.69) | <0.001 |
| Total cesarean, multipara | 0.00 | −0.51 (−0.71 to −0.31) | −0.26 (−0.55 to 0.02) | −0.25 (−0.59 to 0.1) | 0.15 |
| Cesarean before onset of labor | 0.21 | −0.60 (−0.77 to −0.43) | 0.04 (−0.10 to 0.19) | −0.64 (−0.86 to −0.42) | <0.001 |
| Cesarean intra partum | 0.45 | −0.27 (−0.49- to −0.05) | −0.16 (−0.26 to −0.06) | 0.11 (−0.34 to 0.12) | 0.34 |

annual reduction in the post-trend period of −0.60% annually, whereas the control has a non-significant increase of 0.04% annually. With this, HSJ reduced the CS before the onset of labor by −0.64% annually compared to the control group (p<0.000).

## Discussion

Few studies have demonstrated clinical interventions that effectively reduce CS with long-term impact. Our study is the first Danish study to document a significant and sustained reduction in CS from 21.1% to 12.0%, and without compromising maternal or fetal health.

In the neighboring Nordic country of Sweden, a study has examined the impact of a 9-component organisational and cultural change in obstetric care, targeting Robson Group 1 (nulliparous women with a single cephalic pregnancy at term in spontaneous labor) [30]. Results showed a similar relative reduction in CS as HSJ (45.0% vs. 43.1%) [30,31], and the Swedish hospital´s CS rate has remained persistently low [31,32]. What unites the two initiatives is the ongoing focus on the shared belief in safe vaginal births, interdisciplinary teamwork, skills training, and daily conferences to discuss cases of CS. Both interventions showed no negative effect on neonatal outcomes. Differences between the two initiatives include HSJ's inclusion of all pregnancies except Robson 9 + 10 (transverse/oblique lie and premature births) and the additional focus on components aimed at subsequent pregnancies after traumatic births or CS (S1 File).

In Europe, a Slovakian study by Zahumensky et al., developed a multicomponent intervention that effectively reduced the CS rate, but only monitored for two years after the intervention [33]. The CS rate declined from 33.7% to 22.4% with no negative neonatal impact. Compared to HSJ, the intervention had a different offset, where the main contribution to the decline was a restriction of stakeholders' and private obstetricians' access to order non-obstetric CS, and the implementation of a senior clinical management team to approve elective CS [33]. This highlights how the organisation of birth care and cultural attitudes can influence overall CS rates. Several components of the HSJ strategy, such as staff training, audit, feedback, and evidence-based practices, were also implemented in the Slovakian intervention. Efforts were made to empower midwives to act independently during normal vaginal births [33]. This contrasts with Danish clinical practice, where midwives have been authorised since 1714 to manage uncomplicated births independently [34]. Notably, the Slovakian endpoint of 22.4% remains higher than the initial CS rate at HSJ (21.1%), suggesting other cultural and systemic differences between the two settings [35].

Outside Europe, Chaillet et al performed a large-scale randomised trial involving 32 hospitals in Quebec, Canada [18]. A multicomponent intervention was implemented over 1½ years, with outcomes assessed one year later. The CS rate declined significantly, though modestly by 0.7% overall (a relative reduction of 3.0%) [18]. Like the HSJ initiative, the Quebec model included monitoring data, audits, feedback, and best practice implementation. However, the organisational structure differed: Quebec relied on a designated audit committee conducting quarterly feedback and staff education. In contrast to Quebec, the HSJ initiative followed a bottom-up approach, initiated by midwives and obstetricians within the unit and later supported by management. HSJ audits were conducted daily during morning conferences, fostering continuous learning from real-time clinical cases.

The interventions discussed above are all executed in high-income settings. Globally, CS rates are steadily rising, however, with notable regional variation. The highest rates are found in middle-income countries, while the lowest are in low-income countries [36,37]. Rates range from 42.8% in Latin America to just 5.0% in Sub-Saharan Africa (2021) [1]. This disparity reflects both overuse and underuse of CS. Nevertheless, the sustained and unprecedented universal rise in CS rates is a major public health concern. According to WHO, the increase in CS rates is driven by complex interplay of factors, which makes designing and implementing effective interventions challenging. To address this, the WHO recommends universal strategies such as evidence-based guidelines, mandatory second opinions, statistical monitoring, regular audits with timely feedback, and a collaborative midwifery-obstetric care models [4]. These recommendations are implemented in the HSJ model. As global health promotion best practices in reducing CS rates can be implemented through non-medical interventions, it is recommended to increase extensively on these new public health interventions [14,15]

Alternatives to the health complications assigned to performing CS could be an increased use of instrumental birth. In Denmark, the instrumental delivery rate has declined as CS rates have increased [38,39]. A review on CS-rates in low- and middle-income countries suggests that many second-stage CS could be safely avoided through instrumental births, if staff are adequately trained [36]. However, in high-quality care settings, our study and the Blomberg et al study from Sweden [30] have demonstrated significant reductions in CS rates without a corresponding increase in instrumental births. Notably, the Swedish study (Robson Group 1) even reported a significant decrease in instrumental births [30,31]. Both instrumental delivery and emergency CS are associated with traumatic birth experiences, which may negatively affect maternal health [40,41], bonding with the newborn [42], and future fertility intentions [43]. The multicomponent intervention at HSJ managed to reduce CS rates without increasing instrumental births.

As presented above, there are a number of cultural and structural differences between the HSJ initiative and comparable CS reduction initiatives in other countries. Cultural norms, systemic power dynamics and personal beliefs have been documented through several qualitative and mixed-method studies to impact the CS rates significantly [44]. These studies also emphasize that the greatest barriers to reducing CS often arise when implementation efforts clash with existing systems and cultural expectations [13,44–47]. These beliefs were influenced by professional agreement and disagreement among obstetricians and midwives [48]. Components aiming to enhance interdisciplinary cooperation may be important facilitators for the initiative's implementation and sustainability, as recommended by, e.g., WHO [4,49]. A systematic qualitative assessment of promoters for reducing CS rates has shown that professional teamwork between healthcare providers is pivotal for tackling unnecessary CSs [44]. The highest CS rates were among institutions reporting challenges in achieving collaborative work between midwives and obstetricians, and non-collaboration in the organisational culture and policy documents [44]. These findings are consistent with the HSJ initiative, where collaborative teamwork between obstetricians and midwives was actively supported through clinical skills training, supervision, daily joint obstetric conferences, and the participation of obstetricians in normal births (S1 File).

Supporting leadership and time for cultural changes may be pivotal for changes in contemporary maternity care practices. Braithwaite et al. argue that hands-on approaches by leaders play an essential role in adopting innovations [50]. Also, sufficient time is required to create adequate information flow and feedback and observe an intervention's impact [51]. The fact that the twelve components of the HSJ intervention were implemented over nine years may allow the

necessary time for such processes to take place. As demonstrated in Fig 2, the CS rate at HSJ decreased throughout the implementation period, suggesting that components implemented early *and* late contributed to this decrease in CS.

Seven years after the HSJ finished implementation of the initiative, the CS rate is still low (14.2% in 2024) [19], suggesting a high sustainability of the initiative. Kingdon et al. argue that for practice changes to be persistent, key stakeholders must be involved in both the development and implementation of an intervention, as resistance often originates from a feeling of exclusion, resulting in a lack of ownership [44]. In the HSJ initiative, there has been interdisciplinary involvement in both the initiative's development and implementation phases [52].

Finally, so far, most qualitative studies on CS interventions have focused on maternity care provider perspectives [14,13]. While women's perspectives on CS, e.g., regarding inclusion in decision-making processes, giving informed consent, and fear of vaginal childbirth, are well examined, the evidence based on how women respond to different CS intervention components is currently limited [4,14,44,53]. Future research is needed on the acceptability and satisfaction of such interventions among users. In the current study, such outcomes were not available. However, a national satisfaction survey in 2017 showed that women at HSJ generally scored more satisfied with antenatal care and had similar scores as the national average regarding overall satisfaction with birth [54].

The study design has both strengths and limitations. The strength of the current study is the remarkable reduction in CS rates combined with the sustainability of the intervention. The Swedish study by Blomberg et al. has the same approach, and the sustainability and significance resemble results from HSJ [30,31]. Another strength is the study's design. Studies monitoring new interventions to reduce CS rates are often conducted as pre- and post-interventions studies without a control group and without accounting for the underlying trend [30,33] or as randomised designs running over a short period [18]. This study's strength is in comparing the use of ITSA with a control group (CITSA) as this design provides a robust quasi-experimental alternative to the randomised design [55]. Moreover, by using multiple pre- and post-intervention observations, we account for the underlying trends. As the characteristics of the population tend to change slowly over time, confounding factors are rarely a problem [26,27]. Using a control group strengthens the results by excluding confounding due to time-dependent factors affecting HSJ and controls [25–27]. Furthermore, data were prospectively collected from health personnel unaware of the purpose of the study.

A limitation of the study is the absence of a few relevant variables. HSJ provides merged and thus inappropriate data for NICU>24 hours from 2013. Thereby, we restricted the post-intervention period regarding NICU to 2008–2012. Further, our collaborating NGO organisation requested data on the length of breastfeeding and maternal satisfaction. Both measures were either inadequately collected or absent in the Danish registers. Postpartum haemorrhage (PPH) was not routinely collected before 2011. However, on publicly available health data, HSJ had post-intervention a lower risk of PPH > 1000 ml. than the national average (2017: 5,6% vs. 7,0%) [38]. Compared to other studies, a limitation is that the changes at HSJ were conducted incrementally and not under any formal randomised design [18,33]. Finally, although the results suggest a causal relationship between the overall HSJ initiative and the CS rate, we cannot assess the effectiveness of the individual components of this initiative. A qualitative process evaluation of the initiative could have provided valuable insights into the compliance with and acceptability of the intervention components among obstetric staff and women (e.g., staff adherence and women´s experience).

## Conclusion

This study shows the possible impact of a structured cross-sectional effort to reduce cesarean section at a Danish hospital. During nine years, a relative 43.0% decrease in cesarean section rate was accomplished. No adverse effects were observed for maternal/neonatal outcomes (instrumental birth, uterine rupture, NICU admission, or fetal death), supporting the intervention's safety. The intervention demonstrates the feasibility and effectiveness of this interdisciplinary structured change in culture and clinical practice. If this complex intervention can be implemented elsewhere, it may have a huge potential positive impact on maternal and fetal health.

## Supporting information

**S1 File. The stepwise initiative.**
(DOCX)

**S1 Table. Absolute numbers among CS groups.**
(RTF)

**S1 Fig. HSJ and controls stratified by timing for CS and parity.**
(RTF)

**S2 Fig. CS according to modified Robson's Ten Group Classification System.**
(DOCX)

**S3 Fig. Change in maternal characteristics during 2003–2017, HSJ and control.**
(RTF)

**S2 File. STROBE_cohort Cesarean.**
(DOCX)

## Acknowledgments

We are grateful for the help provided by the HSJ hospital regarding data and historical knowledge. Further, we acknowledge the fruitful collaboration with the "Parenthood and Childbirth" organisation in the initial planning of the study.

## Author contributions

**Conceptualization:** Eva Rydahl, Kamilla G Nielsen, Ole Olsen, Helle Johnsen.

**Data curation:** Eva Rydahl.

**Formal analysis:** Eva Rydahl, Ole Olsen, Helle Johnsen.

**Investigation:** Kamilla G Nielsen.

**Methodology:** Eva Rydahl, Kamilla G Nielsen, Ole Olsen, Helle Johnsen.

**Visualization:** Eva Rydahl.

**Writing – original draft:** Eva Rydahl, Kamilla G Nielsen, Ole Olsen, Amalie L Henningsen, Helle Johnsen.

**Writing – review & editing:** Eva Rydahl, Kamilla G Nielsen, Ole Olsen, Amalie L Henningsen, Helle Johnsen.

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
