## [Decision Letter · Decision Letter 0]

2 Jul 2025

Dear Dr. Rydahl,

We look forward to receiving your revised manuscript.

Kind regards,

Shalik Ram Dhital, PhD

Academic Editor

PLOS ONE

Journal Requirements:

3. Thank you for uploading your study's underlying data set. Unfortunately, the repository you have noted in your Data Availability statement does not qualify as an acceptable data repository according to PLOS's standards.

4. Please amend your manuscript to include your abstract after the title page.

Reviewers' comments:

Reviewer's Responses to Questions

**Comments to the Author**

1. Is the manuscript technically sound, and do the data support the conclusions?

Reviewer #1: Yes

Reviewer #2: Yes

2. Has the statistical analysis been performed appropriately and rigorously?

Reviewer #1: Yes

Reviewer #2: Yes

3. Have the authors made all data underlying the findings in their manuscript fully available?

Reviewer #1: Yes

Reviewer #2: Yes

4. Is the manuscript presented in an intelligible fashion and written in standard English?

Reviewer #1: Yes

Reviewer #2: Yes

Reviewer #1: Article Review: Impact of an Interdisciplinary Intervention on Cesarean Delivery Rates

Summary

This study evaluates the effect of an interdisciplinary intervention on the rate of cesarean deliveries. The main finding is that the implementation of the program resulted in a 40% reduction in cesarean delivery rates.

Comments

1. Overall Quality

o The study is well-written, with high-quality data and robust statistical analysis. It is a strong candidate for publication, as it provides valuable insights into reducing cesarean delivery rates.

2. Key Findings

o The observed 40% reduction in cesarean delivery rates is remarkable, particularly given that the intervention primarily involved education of both patients and staff. The authors’ focus on educational strategies highlights a practical and potentially scalable approach to addressing high cesarean delivery rates.

3. Discussion Point: Instrumental Delivery

o One interesting observation is that the rate of instrumental deliveries remained stable while cesarean rates decreased. This finding raises an important question: could an increase in the use of instrumental delivery contribute to a further reduction in cesarean delivery rates?

o It would be beneficial for the authors to address this question in the ‘Discussion’ section by exploring the potential relationship between instrumental delivery practices and cesarean rates. Additionally, including a discussion of any barriers to the use of instrumental delivery within the studied population could provide a deeper understanding of how to optimize the intervention’s outcomes.

Recommendation:

Minor Revision

Reviewer #2: How structured cultural changes can reduce caesarean section rate in a Danish tertiary hospital; a controlled interrupted time series analysis.

Thank you as you invited me to review this manuscript. Please see my comments as follows:

Abstract

This is not clear that what strategy was taken in indexed hospitals to reduce the rate of cesarean compared to the control hospitals.

Introduction

1. readers like to know about the situation of hospitals before implanting new strategies, e.g. are midwives responsible for care during labor and delivery, and what are the responsibilities of obstetricians?

Methods

1. What authors mean about the hospital has six delivery rooms? Are they LDR (labor, delivery, and recovery room), and how was the ratio of midwives to pregnant women?

2. Intervention Description:

a. The intervention ("stepwise initiative to reduce CS rates") is vague. Briefly summarize key components (e.g., guidelines, staff training, audit feedback) in 1–2 sentences.

b. Clarify why 2008 was chosen as the rollout year if implementation was incremental.

3. Control Group Selection:

a. The criteria for control hospitals are logical, but the incremental filtering (n=20 → n=1) could be streamlined for readability. Consider a table for exclusion steps.

b. Justify merging Holbaek and Hjoerring into one control group (e.g., similarity in baseline CS rates?).

4. Missing data

a. The 10% missing BMI data (2003) and NICU admittance changes (2013 onward) are appropriately addressed, but consider sensitivity analyses (e.g., multiple imputation for BMI).

b. For variables with unsystematic missingness (e.g., breastfeeding), explicitly state if they were excluded from all analyses.

5. Confounding:

a. While CITSA accounts for national trends, discuss other potential confounders (e.g., changes in maternal age, obesity rates) and whether they were adjusted for.

b. Mention if parallel trends were tested pre-intervention (critical for ITSA validity).

6. Outcome Definitions:

a. Specify how "foetal death" was defined (stillbirths ≥22 weeks?).

b. Clarify if Apgar scores were categorized (e.g., <7 at 5 minutes) or analyzed continuously.

Results

Baseline Differences (Lines 272–279):

• Use consistent formatting for percentages (e.g., "1.8%" instead of "1,8%").

• Clarify if foetal death difference is statistically significant (add p-value or confidence interval).

• Figure 1 Description (Lines 281–293):

o Specify if the "stable 18% CS rate" refers to the pre-intervention period (2003–2007) or entire timeframe.

o Highlight the 12% CS rate at HSJ (2017) earlier—this is a key finding.

• Time Series Results (Lines 294–331):

o Single-group ITSA (Table 2):

Clarify if "elective/subacute CS" (Line 302) aligns with "CS before onset of labour" (Line 328) or if these are separate categories.

Use consistent terminology (e.g., "elective" vs. "before onset of labour").

o CITSA (Table 3):

Emphasize the −0.75% annual extra reduction at HSJ (Line 324) as a headline result.

Explicitly state if the "Danish awareness" (Line 322) is hypothetical or based on external data (e.g., policy changes).

• Effect Sizes:

o Report confidence intervals (e.g., "−0.87% [95% CI: −1.2, −0.5]") alongside p-values for key trends.

o For non-significant outcomes (e.g., uterine rupture), note if trends were stable or underpowered.

• Autocorrelation/Model Fit:

o Briefly mention if Newey-West adjustments or lag terms were applied (critical for ITSA validity).

• Your results are compelling and methodologically sound. Streamlining terminology, emphasizing effect sizes with CIs, and improving narrative flow will enhance clarity. The dramatic CS reduction at HSJ (−0.87% annually) deserves prominence—consider leading with this finding.

Discussion

1. Principal Findings (Lines 334–343)

• Clarify the "national trend":

o Specify whether the modest control-group decrease (−0.12%) was non-significant (add p-value).

• Negative Outcomes:

o Strengthen the conclusion:

"No adverse effects were observed for maternal/neonatal outcomes (instrumental birth, uterine rupture, NICU admission, or foetal death), supporting the intervention’s safety."

• Limitations:

o Missing Data:

Clarify if NICU >24 hours was excluded entirely post-2013 or analyzed partially.

Cite the public PPH data [31] earlier to contextualize its absence.

o Qualitative Gap:

Highlight how future process evaluations could address this (e.g., staff adherence, women’s experiences).

**Do you want your identity to be public for this peer review?** For information about this choice, including consent withdrawal, please see our Privacy Policy

Reviewer #1: **Yes: ** Ioannis Alagkiozidis

Reviewer #2: **Yes: ** Parvin Abedi

---

## [Author Response · Author response to Decision Letter 1]

26 Aug 2025

Response to Reviewers. A MORE READABLE VERSION IN ATTACHED FILE

PONE-D-24-53886

How structured cultural changes can reduce cesarean section rate in a Danish tertiary hospital; a controlled interrupted time series analysis.

PLOS ONE

Dear Editor-in-Chief

We hereby resubmit the revised manuscript How structured cultural changes can reduce cesarean section rate in a Danish tertiary hospital; a controlled interrupted time series analysis.

Thank you very much for your thorough review of the manuscript and the constructive comments provided. We have thoroughly discussed all comments within the author group and addressed all suggestions. We do hope we have understood and fulfilled the requested amendments satisfactorily.

Concerning the study's underlying data set, we were not aware that Data Availability statement did not qualify to the PLOS's standards. We have made a data repository at Zenodo https://doi.org/10.5281/zenodo.16910955, but have not published it yet, as we were unsure it could compromise the safety of our results. We will change the Data Availability statement when resubmit the study.

Editor´s comment: Normally discussion does not include significant results and percentages. This presentation seems like result.

Please write your discussion with out subheading and focus your discussion on main findings in words, compare of your study findings with relevant previous studies in Denmark, and Europe. You can compare your study findings with other Developed countries like the USA, Australia and so on. Few examples can take from low income countries. You must give reason of any similarities and differential.

Response from the authors:

Thank you for your thorough and valuable comments. We have addressed all these. In the uploaded version, we have removed your comments but retained track changes throughout the document to highlight the revisions made.

We have read your comment regarding the discussion, as you suggested, opting for a change in the paraphrases. Thus, we have changed the structure in the discussion section. We have:

1) Removed the initial section, which included results

2) Removed subtitles

3) Changed the structure “geographically” by comparing the HSJ intervention to other intervention studies from the Nordic, the European and other high-income countries in that order. Furthermore, similarities and differences are discussed as you requested.

4) According to a reviewer comment, we have included a discussion of other methods to reduce CS, more specifically use of instrumental birth as a CS preventative measure.

5) We have also discussed other studies elaborating on why or why not cultural changes may succeed.

6) The strength and limitation paraphrase has been moved to the last part of the discussion.

We hope this aligns with your requests, and we believe our amendments have enhanced the overall quality and structure of our paper. However, if you feel further improvements are needed, please do not hesitate to inform us. For some reason, the track changes in the discussion section disappeared when sending it between authors. We tried but were unable to recreate it with track changes.

Reviewer Comments:

Below, we provide an overview of the amendments made in response to the reviewers’ constructive feedback. A point-by-point response to each comment is included. We respond by marking the text in red, whereas reviewers' comments appear in black font. “Line” refers to the clean version without visible track changes.

Reviewer #1

First of all, we thank you for your comments and relevant inputs for the discussion. We hope that our response and the text changes fulfil your thoughts. Otherwise, please let us know.

Comments to the Author:

Discussion Point: Instrumental Delivery

o One interesting observation is that the rate of instrumental deliveries remained stable while cesarean rates decreased. This finding raises an important question: could an increase in the use of instrumental delivery contribute to a further reduction in cesarean delivery rates?

o It would be beneficial for the authors to address this question in the ‘Discussion’ section by exploring the potential relationship between instrumental delivery practices and cesarean rates. Additionally, including a discussion of any barriers to the use of instrumental delivery within the studied population could provide a deeper understanding of how to optimize the intervention’s outcomes.

Response from the authors:

We agree that instrumental birth may be an accessible way to reduce high cesarean rates, and in some settings, this could be beneficial and decrease morbidity. Especially in settings where CS cannot be performed safely. However, instrumental birth is also associated with negative or traumatic birth experiences, so pros and cons should be carefully considered. The HSJ hospital had an interest in reducing the CS rate without increasing the instrumental birth rate and not compromising birthing experiences. We have added a section to the discussion, especially regarding the use of instrumental birth (Line 440-451)

Reviewer #2

Thank you for your time and the effort you put into our text. Your comments and findings regarding errors and inconsistencies were helpful and improved the overall impression of the text. We have replied to your comments and provided the new text in response below.

Abstract

Comments to the Author:

This is not clear that what strategy was taken in indexed hospitals to reduce the rate of cesarean compared to the control hospitals.

Response from the authors:

We have changed the introducing text, line 18-22, that we hope- in few words- will be more informative to the reader: “This study aims to evaluate the impact of a 12-step initiative implemented by a Danish tertiary hospital, targeting organisational structures, healthcare personnel, and the birthing population, on reducing cesarean section rates, compared to control hospitals that did not adopt a similar approach”.

Introduction

Comments to the Author: readers like to know about the situation of hospitals before implanting new strategies, e.g. are midwives responsible for care during labor and delivery, and what are the responsibilities of obstetricians?

Response from the authors:

We have reformulated the sentence in line 92-95 to be more informative about the relation between midwives and obstetricians, which we hope will clarify the issue: “Normal pregnancies and childbirth are managed independently by authorized midwives. In case of complications, obstetricians take over responsibility, working in close collaboration with the midwife”

Methods

Comments to the Author:

1. What authors mean about the hospital has six delivery rooms? Are they LDR (labor, delivery, and recovery room), and how was the ratio of midwives to pregnant women?

Response from the authors:

We have reformulated the “delivery room” to “birthing room” throughout the text, which we think rightly covers that only women in active labor and birth attend these rooms. In clinical practice, women are transferred to a recovery room after 2-3 hours and many are leaving the hospital within 4-6 hours.

We contacted the leading midwife at HSJ, but she had no data of any midwifes to pregnancy ratio. In Denmark it is a quality assessment how many have 1:1 contact from active labor (usually 6 cm).

Intervention Description:

Comments to the Author:

a. The intervention ("stepwise initiative to reduce CS rates") is vague. Briefly summarize key components (e.g., guidelines, staff training, audit feedback) in 1–2 sentences.

Response from the authors:

More information has been added: “Between 2008 and 2017, HSJ developed and implemented a strategic initiative aimed at reducing the CS rate. The initiative included 12 different components, like e.g. team training, audit feedback, and birth plans for women after CS or traumatic births (S1_file). “Line 179-181

b. Clarify why 2008 was chosen as the rollout year if implementation was incremental.

Response from the authors:

We have changed the text to explain the year 2008 more clearly:

Between 2008 and 2017, HSJ developed and implemented a strategic initiative aimed at reducing the CS rate. The rollout began in 2008, coinciding with the appointment of a new head of department who was specifically tasked with leading the strategy. For this analysis, 2008 is considered the year of introduction. Line 182-184

Control Group Selection:

Comments to the Author:

a. The criteria for control hospitals are logical, but the incremental filtering (n=20 → n=1) could be streamlined for readability. Consider a table for exclusion steps.

Response from the authors:

We did puzzle with such a table, but aggreed not to add one according to the total number of figures and tables. But if this is thought necessary, we could off course add one.

b. Justify merging Holbaek and Hjoerring into one control group (e.g., similarity in baseline CS rates?).

Response from the authors:

We used filtering to identify organizational criteria rather than focusing on outcome similarities, as our primary analytical interest was in trends over time. This means that initial differences in CS rates between hospitals were less relevant than the overall trajectory. Nonetheless, we reviewed the data and found relatively small variation in CS rates in the baseline year (2003): HSJ 23%, Hjoerring 21%, and Holbaek 24%."

Missing data

Comments to the Author:

a. The 10% missing BMI data (2003) and NICU admittance changes (2013 onward) are appropriately addressed, but consider sensitivity analyses (e.g., multiple imputation for BMI).

Response from the authors:

We considered imputing missing data for BMI and NICU to cover the full study period. However, BMI data was only missing for one year (2003) and excluding that year resulted in minimal missingness for the rest of the period. For NICU, the situation was different: data from four years were missing due to registration issues at HSJ. Given that NICU admission is sensitive to cultural and institutional norms, and that imputation could introduce further bias, we chose not to impute. Instead, we relied on more robust indicators of neonatal wellbeing, such as low Apgar scores and fetal death rates.

b. For variables with unsystematic missingness (e.g., breastfeeding), explicitly state if they were excluded from all analyses.

Response from the authors:

We changed the text and explicitly stated the exclusion. However, it was only the case on breastfeeding data: “Data on breastfeeding were available from an external health visitor dataset; however, due to a high degree of unsystematic missing data, they were excluded from all analyses” Line 278-280.

Confounding:

Comments to the Author:

a. While CITSA accounts for national trends, discuss other potential confounders (e.g., changes in maternal age, obesity rates) and whether they were adjusted for.

Response from the authors:

We added more information to the control group section:

“The assumption is that the population characteristics change slowly over time (e.g. maternal age, obesity rates, parity). Thereby, the pre-intervention and the intervention population characteristics presumably are unaltered.” Line 191-193.

Further, we added an introduction to time series in the results section: “Assuming gradual changes in population characteristics over time, ITSA does not inherently adjust for potential confounders. Albeit including a control group (CITSA) account for confounding by co-interventions (e.g., national policy changes on CS rates) [23]. Population-level trends in nulliparity, maternal age >35 years, BMI >30, and smoking are shown in S5 Fig., which reveal the same time trends observed between HSJ and the control hospitals”. Line 346-351

b. Mention if parallel trends were tested pre-intervention (critical for ITSA validity).

Response from the authors:

Parallel trends were tested pre-intervention. The trends pre-intervention are shown in fig 3, and the differences in trends pre-intervention is presented in Table 3. Here the overall difference in trends is p=0.10. For simplicity, we limited Table 3 to show the slopes post-intervention, but think that the Fig 3, the p-value of difference in trends and the text in line 373-374 will be sufficient information.

Outcome Definitions:

Comments to the Author:

a. Specify how "foetal death" was defined (stillbirths ≥22 weeks?).

Response from the authors:

As premature births and births without gestational weeks were excluded from the sample, only fetal deaths after 37+0 is included. For clarity, this information is added in line 247.

b. Clarify if Apgar scores were categorized (e.g., <7 at 5 minutes) or analyzed continuously.

Response from the authors:

Apgar score were categorized, and we have clarified this to “Apgar Score <7/ 5 minutes” in line 230, and the dichotomous outcome is reflected in the results section: Table 2 and lines 362-365.

Results

Comments to the Author:

Baseline Differences (Lines 272–279):

• Use consistent formatting for percentages (e.g., "1.8%" instead of "1,8%").

Response from the authors:

Thanks- this has been corrected.

• Clarify if fetal death difference is statistically significant (add p-value or confidence interval).

Response from the authors:

We have added the p-value for NICU and fetal death in lines 364-365, and it is also presented in Table 2.

• Figure 1 Description (Lines 281–293):

o Specify if the "stable 18% CS rate" refers to the pre-intervention period (2003–2007) or entire timeframe.

Response from the authors:

The sentence has been rewritten to specify the full timespan:

“Denmark CS rate remained relatively stable, fluctuating between 19% and 22% from 2003 onward 2017. However, during the pre-intervention period from 2007 to 2013, a gradual increase was observed, with the rate rising from 18% to 20%. The CS rate rose at both HSJ and the control hospitals during this period. After 2008, the control group increased their CS rate to approximately 22-23%, whereas HSJ decreased to 12% annually (2017)”. Line 327 -332.

o Highlight the 12% CS rate at HSJ (2017) earlier—this is a key finding.

Response from the authors:

Thank you for highlighting this. The main points could easily disappear in the sum of results. We added a brief introduction to the result section, highlighting the key finding:

“The key outcome of this 12-point initiative was that HSJ’s 12-point initiative led to a sustained reduction in the CS rate from 21.1% to 12.0%, without compromising maternal or fetal health. While both HSJ and the control hospital showed declining CS rates after 2008, the intervention yielded an additional annual reduction of 0.75% at HSJ. The following section presents descriptive statistics, followed by time series analyses, both single and with a control group” Line 306-310

Further we added this again in the first sentence of the discussion.

• Time Series Results (Lines 294–331):

o Single-group ITSA (Table 2):

Clarify if "elective/subacute CS" (Line 302) aligns with "CS before onset of labour"

(Line 328) or if these are separate categories.

Use consistent terminology (e.g., "elective" vs. "before onset of labour").

o CITSA (Table 3):

Response from the authors:

Thank you- the inconsistency has been corrected, and only “before onset of labor” is now used throughout the article.

Emphasize the −0.75% annual extra reduction at HSJ (Line 324) as a headline result.

Response from the authors:

See the above intro to results, where this finding is also introduced.

Explicitly state if the "Danish awareness" (Line 322) is hypothetical or based on external data (e.g., policy changes).

Response from the authors:

Thanks! - The sentence has been rewritten for clarification: “The overall decline suggests a growing national awareness in Denmark of the need to stabilize or reduce CS rates, even without formal policy changes”. Line 376-377

• Effect Sizes:

o Report confidence intervals (e.g., "−0.87% [95% CI: −1.2, −0.5]") alongside p-values for key trends.

Response from the authors:

This has been corrected throughout the paraphrase. Line 373-386

o For non-significant outcomes (e.g., uterine rupture), note if t

---

## [Editor Report · Decision Letter 1]

31 Aug 2025

Dear Dr. Eva

Thank you for addressing most of the comments received from reviewers and resubmitting your manuscript to PLOS ONE. After careful consideration, we feel that it has merit but does not fully meet PLOS ONE’s publication criteria as it currently stands. Therefore, we invite you to submit a revised version of the manuscript that addresses the points raised during the review process.

We look forward to receiving your revised manuscript.

Kind regards,

Shalik Ram Dhital, PhD

Academic Editor

PLOS ONE
---

## [Author Response · Author response to Decision Letter 2]

8 Sep 2025

Response to Reviewers

PONE-D-24-53886

How structured cultural changes can reduce cesarean section rate in a Danish tertiary hospital; a controlled interrupted time series analysis.

PLOS ONE

Dear Editor-in-Chief

We hereby resubmit the revised manuscript How structured cultural changes can reduce cesarean section rate in a Danish tertiary hospital; a controlled interrupted time series analysis.

Thank you very much for your thorough review of the manuscript and the constructive comments provided. We have thoroughly discussed all comments within the author group and addressed all suggestions. We do hope we have understood and fulfilled the requested amendments satisfactorily.

Concerning the study's underlying data set, we were not aware that Data Availability statement did not qualify to the PLOS's standards. We have made a data repository at Zenodo https://doi.org/10.5281/zenodo.16910955, but have not published it yet, as we were unsure it could compromise the safety of our results. We will change the Data Availability statement when resubmit the study.

Editor´s comment: Normally discussion does not include significant results and percentages. This presentation seems like result.

Please write your discussion with out subheading and focus your discussion on main findings in words, compare of your study findings with relevant previous studies in Denmark, and Europe. You can compare your study findings with other Developed countries like the USA, Australia and so on. Few examples can take from low income countries. You must give reason of any similarities and differential.

Response from the authors:

Thank you for your thorough and valuable comments. We have addressed all these. In the uploaded version, we have removed most comments but retained track changes throughout the document to highlight the revisions made.

We have read your comment regarding the discussion, as you suggested, opting for a change in the paraphrases. Thus, we have changed the structure in the discussion section. We have:

1) Removed the initial section, which included results

2) Removed subtitles

3) Changed the structure “geographically” by comparing the HSJ intervention to other intervention studies from the Nordic, the European and other high-income countries in that order. Furthermore, similarities and differences are discussed as you requested.

4) According to a reviewer comment, we have included a discussion of other methods to reduce CS, more specifically use of instrumental birth as a CS preventative measure.

5) We have also discussed other studies elaborating on why or why not cultural changes may succeed.

6) The strength and limitation paraphrase has been moved to the last part of the discussion.

2.revision 1 September

7) Removed “:” after subtitles

8) A flowchart has been implemented in Fig 1.

9) Aligned the text with 1 decimal after all numbers throughout the text.

10) We kept the comments in the revised Manuscript with track changes.

Authors suggest a sentence and references:

Can you say some examples of clinical and non clinical strategies? For examples non clinical like health promotion, health education, nutrition promotion, regular check up, financial management, delay management and quality of maternal care etc. please add this reference

We have exemplified clinical and non-clinical interventions in lines 73-75

Authors suggest a sentence and references:

Global health promotion best practices in maternal and neonatal health care have to be successfully implemented in reducing CS rates through non-medical interventions, so the best way is to increase extensively new public health interventions [16,54]

We recognize this valuable input. However, we think it is better placed in the paraphrase regarding global initiatives, and moved it to lines 441-444. We hope that editor agrees with us.

We hope this aligns with your requests, and we believe our amendments have enhanced the overall quality and structure of our paper. However, if you feel further improvements are needed, please do not hesitate to inform us. After the Reviewers comments and reply follows a section on Author contribution and changes in references.

Reviewer Comments:

Below, we provide an overview of the amendments made in response to the reviewers’ constructive feedback. A point-by-point response to each comment is included. We respond by marking the text in red, whereas reviewers' comments appear in black font. “Line” refers to the clean version without visible track changes.

Reviewer #1

First of all, we thank you for your comments and relevant inputs for the discussion. We hope that our response and the text changes fulfil your thoughts. Otherwise, please let us know.

Comments to the Author:

Discussion Point: Instrumental Delivery

o One interesting observation is that the rate of instrumental deliveries remained stable while cesarean rates decreased. This finding raises an important question: could an increase in the use of instrumental delivery contribute to a further reduction in cesarean delivery rates?

o It would be beneficial for the authors to address this question in the ‘Discussion’ section by exploring the potential relationship between instrumental delivery practices and cesarean rates. Additionally, including a discussion of any barriers to the use of instrumental delivery within the studied population could provide a deeper understanding of how to optimize the intervention’s outcomes.

Response from the authors:

We agree that instrumental birth may be an accessible way to reduce high cesarean rates, and in some settings, this could be beneficial and decrease morbidity. Especially in settings where CS cannot be performed safely. However, instrumental birth is also associated with negative or traumatic birth experiences, so pros and cons should be carefully considered. The HSJ hospital had an interest in reducing the CS rate without increasing the instrumental birth rate and not compromising birthing experiences. We have added a section to the discussion, especially regarding the use of instrumental birth. Lines 437-448

Reviewer #2

Thank you for your time and the effort you put into our text. Your comments and findings regarding errors and inconsistencies were helpful and improved the overall impression of the text. We have replied to your comments and provided the new text in response below.

Abstract

Comments to the Author:

This is not clear that what strategy was taken in indexed hospitals to reduce the rate of cesarean compared to the control hospitals.

Response from the authors:

We have changed the introducing text, lines 17-21, that we hope- in few words- will be more informative to the reader: “This study aims to evaluate the impact of a 12-step initiative implemented by a Danish tertiary hospital, targeting organisational structures, healthcare personnel, and the birthing population, on reducing cesarean section rates, compared to control hospitals that did not adopt a similar approach”.

Introduction

Comments to the Author: readers like to know about the situation of hospitals before implanting new strategies, e.g. are midwives responsible for care during labor and delivery, and what are the responsibilities of obstetricians?

Response from the authors:

We have reformulated the sentence in lines 88-91 to be more informative about the relation between midwives and obstetricians, which we hope will clarify the issue: “Normal pregnancies and childbirth are managed independently by authorized midwives. In case of complications, obstetricians take over responsibility, working in close collaboration with the midwife”

Methods

Comments to the Author:

1. What authors mean about the hospital has six delivery rooms? Are they LDR (labor, delivery, and recovery room), and how was the ratio of midwives to pregnant women?

Response from the authors:

We have reformulated the “delivery room” to “birthing room” throughout the text, which we think rightly covers that only women in active labor and birth attend these rooms. In clinical practice, women are transferred to a recovery room after 2-3 hours and many are leaving the hospital within 4-6 hours.

We contacted the leading midwife at HSJ, but she had no data of any midwifes to pregnancy ratio. In Denmark it is a quality assessment how many have 1:1 contact from active labor (usually 6 cm).

Intervention Description:

Comments to the Author:

a. The intervention ("stepwise initiative to reduce CS rates") is vague. Briefly summarize key components (e.g., guidelines, staff training, audit feedback) in 1–2 sentences.

Response from the authors:

More information has been added: “Between 2008 and 2017, HSJ developed and implemented a strategic initiative aimed at reducing the CS rate. The initiative included 12 different components, like e.g. team training, audit feedback, and birth plans for women after CS or traumatic births (S1_file). “Lines 177-179

b. Clarify why 2008 was chosen as the rollout year if implementation was incremental.

Response from the authors:

We have changed the text to explain the year 2008 more clearly:

Between 2008 and 2017, HSJ developed and implemented a strategic initiative aimed at reducing the CS rate. The rollout began in 2008, coinciding with the appointment of a new head of department who was specifically tasked with leading the strategy. For this analysis, 2008 is considered the year of introduction. Lines 188-182

Control Group Selection:

Comments to the Author:

a. The criteria for control hospitals are logical, but the incremental filtering (n=20 → n=1) could be streamlined for readability. Consider a table for exclusion steps.

Response from the authors:

We did puzzle with such a table, but aggreed not to add one according to the total number of figures and tables. But if this is thought necessary, we could off course add one.

b. Justify merging Holbaek and Hjoerring into one control group (e.g., similarity in baseline CS rates?).

Response from the authors:

We used filtering to identify organizational criteria rather than focusing on outcome similarities, as our primary analytical interest was in trends over time. This means that initial differences in CS rates between hospitals were less relevant than the overall trajectory. Nonetheless, we reviewed the data and found relatively small variation in CS rates in the baseline year (2003): HSJ 23%, Hjoerring 21%, and Holbaek 24%."

Missing data

Comments to the Author:

a. The 10% missing BMI data (2003) and NICU admittance changes (2013 onward) are appropriately addressed, but consider sensitivity analyses (e.g., multiple imputation for BMI).

Response from the authors:

We considered imputing missing data for BMI and NICU to cover the full study period. However, BMI data was only missing for one year (2003) and excluding that year resulted in minimal missingness for the rest of the period. For NICU, the situation was different: data from four years were missing due to registration issues at HSJ. Given that NICU admission is sensitive to cultural and institutional norms, and that imputation could introduce further bias, we chose not to impute. Instead, we relied on more robust indicators of neonatal wellbeing, such as low Apgar scores and fetal death rates.

b. For variables with unsystematic missingness (e.g., breastfeeding), explicitly state if they were excluded from all analyses.

Response from the authors:

We changed the text and explicitly stated the exclusion. However, it was only the case on breastfeeding data: “Data on breastfeeding were available from an external health visitor dataset; however, due to a high degree of unsystematic missing data, they were excluded from all analyses” Lines 274-276.

Confounding:

Comments to the Author:

a. While CITSA accounts for national trends, discuss other potential confounders (e.g., changes in maternal age, obesity rates) and whether they were adjusted for.

Response from the authors:

We added more information to the control group section:

“The assumption is that the population characteristics change slowly over time (e.g. maternal age, obesity rates, parity). Thereby, the pre-intervention and the intervention population characteristics presumably are unaltered.” Lines 189-191.

Further, we added an introduction to time series in the results section: “Assuming gradual changes in population characteristics over time, ITSA does not inherently adjust for potential confounders. Albeit including a control group (CITSA) account for confounding by co-interventions (e.g., national policy changes on CS rates) [23]. Population-level trends in nulliparity, maternal age >35 years, BMI >30, and smoking are shown in S5 Fig., which reveal the same time trends observed between HSJ and the control hospitals”. Lines 339-344

b. Mention if parallel trends were tested pre-intervention (critical for ITSA validity).

Response from the authors:

Parallel trends were tested pre-intervention. The trends pre-intervention are shown in fig 3, and the differences in trends pre-intervention is presented in Table 3. Here the overall difference in trends is p=0.10. For simplicity, we limited Table 3 to show the slopes post-intervention, but think that the Fig 3, the p-value of difference in trends and the text in lines 373-374 will be sufficient information.

Outcome Definitions:

Comments to the Author:

a. Specify how "foetal death" was defined (stillbirths ≥22 weeks?).

Response from the authors:

As premature births and births without gestational weeks were excluded from the sample, only fetal deaths after 37+0 is included. For clarity, this information is added in line 243.

b. Clarify if Apgar scores were categorized (e.g., <7 at 5 minutes) or analyzed continuously.

Response from the authors:

Apgar score were categorized, and we have clarified this to “Apgar Score <7/ 5 minutes” in line 228, and the dichotomous outcome is reflected in the results section: Table 2 and lines 357-359.

Results

Comments to the Author:

Baseline Differences (Lines 272–279):

• Use consistent formatting for percentages (e.g., "1.8%" instead of "1,8%").

Response from the authors:

Thanks- this has been corrected.

• Clarify if fetal death difference is statistically significant (add p-value or confidence interval).

Response from the authors:

We have added the p-value for NICU and fetal death in lines 364-365, and it is also presented in Table 2.

• Figure 1 Description (Lines 281–293):

o Specify if the "stable 18% CS rate" refers to the pre-intervention period (2003–2007) or entire timeframe.

Response from the authors:

The sentence has been rewritten to specify the full timespan:

“Denmark CS rate remained relatively stable, fluctuating between 19% and 22% from 2003 onward 2017. However, during the pre-intervention period from 2007 to 2013, a gradual increase was observed, with the rate rising from 18% to 20%. The CS rate rose at both HSJ and the control hospitals during this period. After 2008, the control group increased their CS rate to approximately 22-23%, whereas HSJ decreased to 12% annually (2017)”. Lines 329-334.

o Highlight the 12% CS rate at HSJ (2017) earlier—this is a key finding.

Response from the authors:

Thank you for highlighting this. The main points could easily disappear in the sum of results. We added a brief introduction to the result section, highlighting the key finding:

“The key outcome of this 12-point initiative was that HSJ’s 12-point initiative led to a sustained reduction in the CS rate from 21.1% to 12.0%, without compromising maternal or fetal health. While both HSJ and the control hospital showed declining CS rates after 2008, the intervention yielded an additional annual reduction of 0.75% at HSJ. The following section presents descriptive statistics, followed by time series analyses, both single and with a control group” Lines 302-306

Further we added this again in the first sentence of the discussion.

• Time Series Results (Lines 294–331):

o Single-group ITSA (Table 2):

Clarify if "elective/subacute CS" (Line 302) aligns with "CS before onset of labour"

(Line 328) or if these are separate categories.

Use consistent terminology (e.g., "elective"

---

## [Editor Report · Decision Letter 2]

25 Sep 2025

Dear Dr. Rydahl,

 The Academic Editor requests some minor revisions to your manuscript (see attached file "Manuscript Cesarean 010925 Track changes__AE_ Sep 19.docx"). Could you please revise the manuscript to carefully address the issues raised?

We look forward to receiving your revised manuscript.

Kind regards,

Steve Zimmerman, PhD

Senior Editor, PLOS One

on behalf of

Shalik Ram Dhital, PhD

Academic Editor

PLOS ONE
---

## [Author Response · Author response to Decision Letter 3]

26 Sep 2025

26.September: Comments from editor has been adressed, and a new version Manuscript 260925 with track changes has been uploaded and a new version of Response to the editor has been uploaded.

---

## [Editor Report · Decision Letter 3]

15 Oct 2025

Dear Dr. Rydahl,

We look forward to receiving your revised manuscript, both clean and with track changes.

Kind regards,

Shalik Ram Dhital, PhD

Academic Editor

PLOS ONE

Journal Requirements:

Additional Editor Comments:

Please follow the PLOS ONE Author Guidelines and strictly consider each component of preparing a manuscript.  I have provided feedback on the Track Changes manuscript. Please address my comments and resubmit after adequate discussion with all authors.

---

## [Author Response · Author response to Decision Letter 4]

21 Oct 2025

21.October: We have adressed addtional comments from editor. Mainly regarding our misunderstanding of how to cite figures in the manuscript. We hope today´s version fits the requirements. Following changes has been made in the new 211025 version:

• In the manuscript, the figures have been cited according to the file text, and the flowchart has been removed from the text and uploaded as a figure.

• The “Supporting information” has been moved to present before “References” as recommended by the editor.

• The section starting line 429 (track change version) has been remodelled to address the editor's comment

• Few other comments on words or punctuation have been addressed

---

## [Editor Report · Decision Letter 4]

27 Oct 2025

How structured cultural changes can reduce caesarean section rate in a Danish tertiary hospital.

PONE-D-24-53886R4

Dear Dr. Eva,

We’re pleased to inform you that your manuscript has been judged scientifically suitable for publication and will be formally accepted for publication once it meets all outstanding technical requirements.

Kind regards,

Shalik Ram Dhital, PhD

Academic Editor

PLOS ONE

---

## [Editor Report · Acceptance letter]

PONE-D-24-53886R4

PLOS ONE

Dear Dr. Rydahl,

I'm pleased to inform you that your manuscript has been deemed suitable for publication in PLOS ONE. Congratulations! Your manuscript is now being handed over to our production team.

Kind regards,

on behalf of

Dr. Shalik Ram Dhital

Academic Editor

PLOS ONE